# Relationship between the Water Quality Elements of Water Bodies and the Hydrometric Parameters: Case Study in Lithuania

**Laima Česonienė [1],\*** , **Daiva Šileikienė [1]** **and Midona Dapkienė [2]**

[1] Institute of Environment and Ecology, Faculty of Forest Science and Ecology, Agriculture Academy, Vytautas Magnus University, Studentų g. 11, LT–53361 Akademija, Kauno raj, Lithuania; daiva.sileikiene@vdu.lt

[2] Institute of Hydraulic Engineering, Faculty of Water and Land Management, Agriculture Academy, Vytautas Magnus University, Studentų g. 11, LT–53361 Akademija, Kauno raj, Lithuania; midona.dapkiene@vdu.lt

\* Correspondence: laima.cesoniene1@vdu.lt; Tel.: +370-37-752224

**Abstract:** The larger and deeper lakes and ponds are, the better the conditions for spontaneous water purification, slower hydrobiological processes and slower accumulation of sediment. The goal of this research was to assess the ecological status of selected Lithuanian lentic water bodies and the impact of morphometric indicators on water quality. Multiple studies were conducted on 29 lakes and 10 ponds located throughout Lithuania in 2014–2018. The study proved that higher maxima and average depths of lakes correlate with lower $P_{total}$, $N_{total}$ yield and macrophyte taxonomic composition values, indicating higher ecological status class. Higher chlorophyll a EQR, ichthyofauna taxonomic composition indicator for Lithuanian fish index LFI and Lithuanian lakes' macroinvertebrate index indicates a higher ecological class. Larger lake areas contain smaller amounts of $P_{total}$ and $N_{total}$, indicating better ecological status class; higher ichthyophane taxonomic composition in LFI, zoobenthos taxonomic composition indicator for Lithuanian lakes' macroinvertebrates index (LLMI) and taxonomic composition of macrophytes MRI indicate better ecological status class. Larger lake areas contain lower chlorophyll a EQR values. Rapid water exchange improves the condition of the lake in addition to nitrogen, phosphorus and chlorophyll a EQR values. The faster the water exchange in the lake is, the lower the $P_{total}$ and $N_{total}$ values; faster water exchange in the lake also means higher chlorophyll a EQR values. However, slower water exchange indicates better ecological status of the macrophytic taxonomic composition of the MRI, the ichthyofauna taxonomic composition and the Lithuanian lakes' macroinvertebrates index indicator of zoobenthos.

**Keywords:** water quality; lake status; depths; volume; sedimentation

## 1. Introduction

Human agricultural activities and their development have an inevitable negative impact on the environment. One of the largest ecological issues today is the intensive anthropogenic activity in throughout the catchment, resulting in eutrophication [1]. Both diffuse pollution and concentrated pollution have a negative impact on surface water bodies. Human activities determine over 90 percent of the annual total nitrogen flow and 78 percent of the total phosphorus flow [2].

Two important directives have been adopted in the EU to mitigate the impact of pollutant emissions on the environment: The Nitrates Directive (1991/696/EC) [3] and the Water Framework Directive (2000/60/EC) [4]. The aim of the implementation of these directives is to protect all water bodies from anthropogenic interference [5].

The European Water Framework Directive (WFD) sets strict environmental quality objectives for surface waters, demands the integrated assessment of environmental pressures and risks failing to meet these objectives [4]. The WFD quality objectives are based on metrics involving four biological quality elements: phytoplankton, aquatic macrophytes, benthic macroinvertebrates, and fish. The chemical parameters (nutrients) are also applied to identify reference conditions and good and very good status [6]. Good and very good ecological status is a state of aquatic fauna and flora living close to natural conditions unaffected by human activity.

A number of studies have considered the ecological status of surface water bodies [7–12].

Eutrophication of surface waters by phosphorus and nitrogen is a major problem that impedes water bodies from meeting the status defined by the WFD. The assessment of the proportion of nitrogen input on the load of water bodies showed that 25 percent of the cases in the area of a surface water body may represent a critical load, leading to an adverse assessment of ecological status in the Vltava river basin (Czech Republic). In many other surface waters, it can, because of the significant load of mineral fertilizers, lead to exceeding the allowable capacity of the water bodies and the risk of not achieving a good status [13].

Monitoring of nutrient losses to surface waters of small agricultural catchments has been carried out in seven Nordic and Baltic countries with the aim of obtaining information on agricultural activities and their impact on surrounding waters [14–16]. The effects of nutrients on the conditions of lakes were studied by Brett and Benjamin [17] and Matysik et al. [18]. Studies of morphometric, physicochemical and vegetation structures, the temporal and seasonal variabilities of hydrochemical properties, nitrogen and phosphorus in bottom sediments and eutrophication problems were carried out for lakes and dam reservoirs in Poland [19–23]. Intense economic activity in lakes and dam reservoir basins had a significant impact on the majority of Lithuanian water body ecosystem, hydrochemical, including changes in water vegetation [24].

It is estimated that 20 to 80 percent of nitrogen is stored in the North Atlantic Ocean [25], 33 percent of nitrogen and 35 percent of phosphorus reside in Estonia [26], and 50 percent of nitrogen resides in Sweden [27]. Lithuanian lakes accumulate 27 to 59 percent of nitrogen and 11 to 31 percent of phosphorus [28], of which 27 to 56 percent of phosphorus is accumulated in southeastern Lithuania [29].

The ecological status assessment of the WFD combines information on several hydromorphological, chemical and biological parameters to acquire a comprehensive picture of the overall status on the function and structure of the ecosystem [6].

A total of 1239 surface water bodies (rivers and lakes) have been defined in Lithuania according to the requirements of the WFD. Experience shows that even though Lithuania implements EU Council directives and normative acts, the preventive actions are not enough to achieve good status of surface water bodies, as many Lithuanian water bodies are still considered to be at risk. Risk pools are those water bodies that may not reach good ecological status by 2027.

There are approximately 2850 lakes in Lithuania, which comprise 1.4 percent of the country's territory. Thirty percent of those lakes or ponds and 60 percent of rivers do not meet good ecological status. In river basin district management plans prepared for the years 2010 to 2015, 30 percent of lakes did not comply with good ecological status, which means that these lakes have been assessed to be at risk, and remedial measures should apply. A total of 790 lakes (the list is regularly updated) have been pre-identified to be at risk [30].

Economic activity was determined to have an impact on 64 percent of Lithuanian surface water bodies. One study showed that approximately 88 percent of water bodies at risk are classified as potentially at risk. The largest number of surface water bodies at risk (75 percent) is caused by point pollution (urban and industrial wastewater), such as agricultural pollution (22 percent of all risk areas). The largest impact is observed in rich soil territories, such as Mūša-Lielupė, Nevėžis, Šešupė, and Bartuva basins and sub-basins, which are the most agriculturally developed places in Lithuania. In particular, polluted surface waters are found in the Joniškis, Pasvalys, Pakruojis (Mūša-Lielupė),

Radviliškis, Kėdainiai, Panevėžys (Nevėžis), Raseiniai (Dubysa), Vilkaviškis (Šešupė) and Skuodas (Bartuva) district municipalities [31].

Agricultural development trends are not favourable from an environmental point of view. Increasing crop area increases fertilization volumes, simultaneously decreasing the number of farm animals, resulting in lower fertilization with animal manure and higher demand for mineral fertilization. The largest aspect of pollution load is in the use of mineral fertilizers because when using mineral fertilisation, the nutrients leaching on the surface water bodies are much higher. In addition to diffused pollution, other factors affecting the status of surface water bodies are hydromorphological changes in surface water bodies caused by land drainage, hydroelectric power stations and river damming, secondary pollution resulting from long-term historical pollution, pollution by hazardous substances and international pollution—pollutants coming from neighbouring countries [31].

The aim of this study is to assess the ecological status of selected Lithuanian lentic water bodies and the impact of morphometric indicators on water quality.

## 2. Materials and Methods

Selected lakes and ponds with different morphometric characteristics were included in the list of Lithuanian water bodies at risk (Minister of Environment of the Republic of Lithuania; on the approved list of at-risk water bodies) [30]. Multiple studies were conducted on 29 lakes and 10 ponds located throughout Lithuania in 2014–2018, each season, four times a year. Lake and pond water samples were taken under the same natural conditions, namely, sunny days. After the collection, all samples were transferred to a laboratory and stored in a refrigerator (5 °C). Water samples are taken according to EN ISO standards: LST EN ISO 5667-14:2016—water quality—Sampling—Part 14 [32].

The following studies were conducted to assess the ecological status of water bodies:

- Research on the indicators of physico-chemical quality elements. The concentrations of total nitrogen ($N_{total}$) and total phosphorus ($P_{total}$) were evaluated. The following analysis methods were applied: total nitrogen ($N_{total}$) was tested according to the method LST EN 13342-2002. Determination of nitrogen—Determination of bound nitrogen (TNb), following oxidation to nitrogen oxides EN 12260:2003 [33]; total phosphorus ($P_{total}$) studies were performed according to LST EN ISO 6878:2004 [34]
- Macrozoobenthos study samples taken from September 2016 to October 2018. Macrozoobenthos studies were conducted using the approved method LAND 57-2003 [35].
- Phytoplankton research. A total of 346 samples were taken. Phytoplankton studies were conducted for the LAND 53-2003 [36].
- Chlorophyll a is determined by the spectrophotometric method LAND 69-2005 [37]. Chlorophyll 'a' EQS was calculated in accordance with the Environmental Regulation of the Republic of Lithuania approved by the Minister of Environment of the Republic of Lithuania on 2005. December 28th Order no. D1-648 (Zinios, 2006, No. 53–123).
- The status of lakes by the reference index of macrophytes MRI principle is assessed by phytobenthos and macrophyte surveys in surface water bodies and ecological status by a macrophyte reference index report, 2016 [38].
- The status of lakes according to the Lithuanian fish index values LFI was evaluated according to the report of 'Ichthyofauna and assessment of the ecological status of Lithuanian rivers and lakes by fish indicators' in 2009–2013 [39].

*Macrozoobenthos* samples in lakes and ponds were taken by using the O'Hare et al. [40] methodology.

The ecological status of water bodies at risk was assessed in accordance with the decision of the Minister of Environment of the Republic of Lithuania on 12 April 2007, by Order no. D1-210 of the Approved Methodology for Determination of the Status of Surface Water Bodies [41].

Correlation and regression were calculated using the computer program STATISTICA 8 [42,43]. The STATISTICA package Nonlinear Estimation was used to determine the correlation coefficients and

to define the relationships between the indicators surveyed [43]. The symbol * indicates that the data were reliable within a probability of 95%. The average and maximum lake depth correlations to the indicator values are calculated by STATISTICA 8, which graphically depicts and performs regression equations, which can be used when predicting changes in indicator values at different depths.

The influence of lake area on indicator values estimated using STATISTICA 8 graphically depicted and calculated regression equations that can be used to predict changes in indicator values if 100 ha of the water area changes.

The sampling points in the studied lakes and ponds are shown in Figure 1.

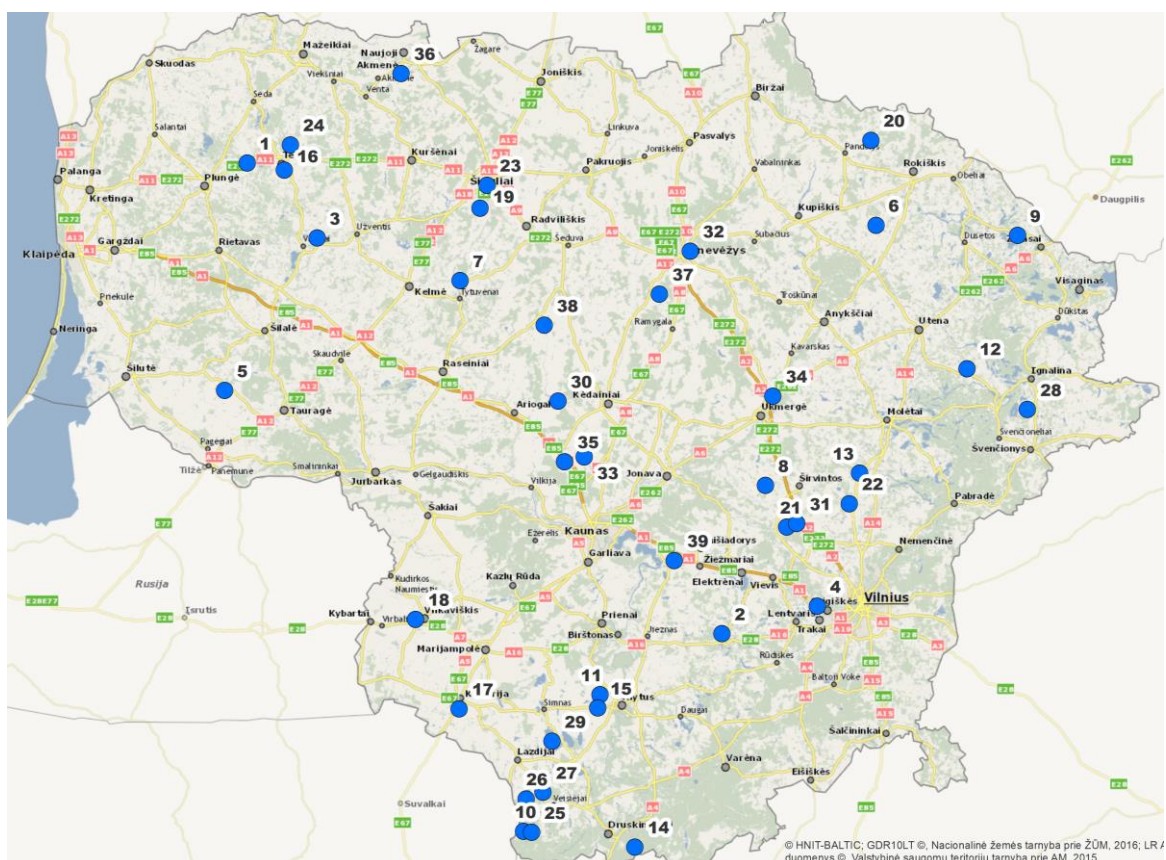

**Figure 1.** Water sampling points in lakes and ponds. Lithuania. Map scale 1:1 255000.

The main morphometric indicators, lake and pond data assessed are presented in Table 1.

**Table 1.** Main morphometric indicators and lake and pond data assessed [44].

| No | Lakes | Average Depth, m | Maximum Depth, m | Water Exchange Per Year | Area, ha | Pool Area, km² |
|----|-------|------------------|------------------|-------------------------|----------|-----------------|
| 1 | Alsėdis | 1.74 | 1.74 | 0.06 | 90.40 | 61.66 |
| 2 | Antakmenis | 5.83 | 12.50 | 1.14 | 84.20 | 17.15 |
| 3 | Biržulis | 0.91 | 0.91 | 0.02 | 114.20 | 137.00 |
| 4 | Didžiulis | 6.00 | 19.00 | 0.14 | 62.10 | 89.96 |
| 5 | Draudenis | 1.51 | 2.50 | 0.24 | 106.50 | 15.07 |
| 6 | Dviragis | 3.07 | 5.39 | 0.72 | 286.60 | 48.15 |
| 7 | Gauštvinis | 5.00 | 5.00 | 0.20 | 124.30 | 154.92 |
| 8 | Gelvanė | 2.39 | 6.10 | 0.54 | 53.00 | 6.31 |
| 9 | Imbradas | 2.00 | 2.00 | 0.43 | 58.70 | 161.78 |
| 10 | Juodas Kauknoris | 4.34 | 13.90 | 0.48 | 58.20 | 20.59 |

**Table 1.** *Cont.*

| No | Lakes | Average Depth, m | Maximum Depth, m | Water Exchange Per Year | Area, ha | Pool Area, km² |
|----|-------|------------------|------------------|-------------------------|----------|----------------|
| 11 | Kavalys | 2.97 | 6.40 | 3.30 | 140.40 | 10.21 |
| 12 | Kemėšys | 4.10 | 4.10 | 0.39 | 53.00 | 9.03 |
| 13 | Kiementas | 3.97 | 7.30 | 0.51 | 98.60 | 26.07 |
| 14 | Latežeris | 1.00 | 9.50 | 0.18 | 86.20 | 14.20 |
| 15 | Luksnėnis | 4.20 | 7.40 | 3.63 | 61.40 | 5.79 |
| 16 | Mastis | 2.60 | 2.60 | 0.54 | 272.20 | 36.81 |
| 17 | Orija | 4.20 | 7.70 | 2.58 | 85.30 | 6.62 |
| 18 | Paežerys | 5.10 | 15.00 | 11.15 | 399.10 | 13.34 |
| 19 | Rėkyva | 2.04 | 2.04 | 7.13 | 1179.20 | 18.07 |
| 20 | Skaistė | 4.93 | 13.10 | 1.79 | 59.00 | 1.81 |
| 21 | Spėra | 1.90 | 2.90 | 0.30 | 80.00 | 21.77 |
| 22 | Širvys | 2.47 | 4.20 | 0.20 | 85.10 | 27.27 |
| 23 | Talkša | 3.46 | 8.70 | 0.45 | 56.20 | 17.82 |
| 24 | Tausalas | 3.34 | 6.10 | 1.89 | 191.20 | 11.99 |
| 25 | Veisiejis | 5.73 | 33.80 | 0.62 | 765.20 | 144.07 |
| 26 | Zapsys | 7.53 | 17.30 | 0.31 | 190.40 | 74.70 |
| 27 | Šlavantas | 11.40 | 28.80 | 2.00 | 65.40 | 187.30 |
| 28 | Kretuonykštis | 1.60 | 3.00 | 0.04 | 65.40 | 85.50 |
| 29 | Dusia | 15.4 | 32.6 | 17.5 | 2334.00 | 107.80 |
| 30 | Angiriai pond | 6.2 | 15.5 | | 264 | 103,515.0 |
| 31 | Bartkuskis pond | 2.5 | 8 | | 52.6 | 23,339.0 |
| 32 | Ekrano gamyklos pond | 2.4 | 6.0 | | 1593 | 106,512.0 |
| 33 | Janušonis pond | 3.5 | 6.8 | | 61.9 | 11,809.0 |
| 34 | Kadrėnai pond | 2.0 | 4.5 | | 107.5 | 22,323.0 |
| 35 | Krivėnai pond | 4.2 | 13.6 | | 67.6 | 10,323.0 |
| 36 | Sablauskai pond | 3.2 | 6.1 | | 125 | 28,586.0 |
| 37 | Stepanioniai pond | 2.8 | 6.5 | | 64.3 | 15,518.0 |
| 38 | Vaitiekūnai pond | 3.6 | 11.0 | | 141.3 | 79,345.0 |
| 39 | Bubliai pond | 4.2 | 13.2 | | 150 | 64,911.0 |

Deepest lakes: Dusia (15.4 m) and Šlavantas (11.4 m); shallowest lake: Biržulis Lake (0.91 m).

## 3. Results

The ecological statuses of lakes and ponds were assessed by the rates of $N_{total}$, $P_{total}$. Macrophytes and chlorophyll a in the water, as presented in Figure 2.

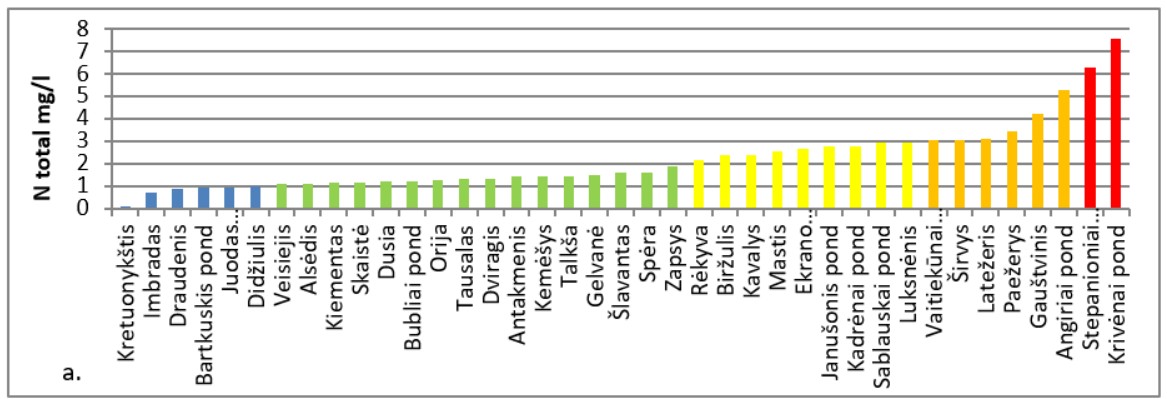

**Figure 2.** *Cont.*

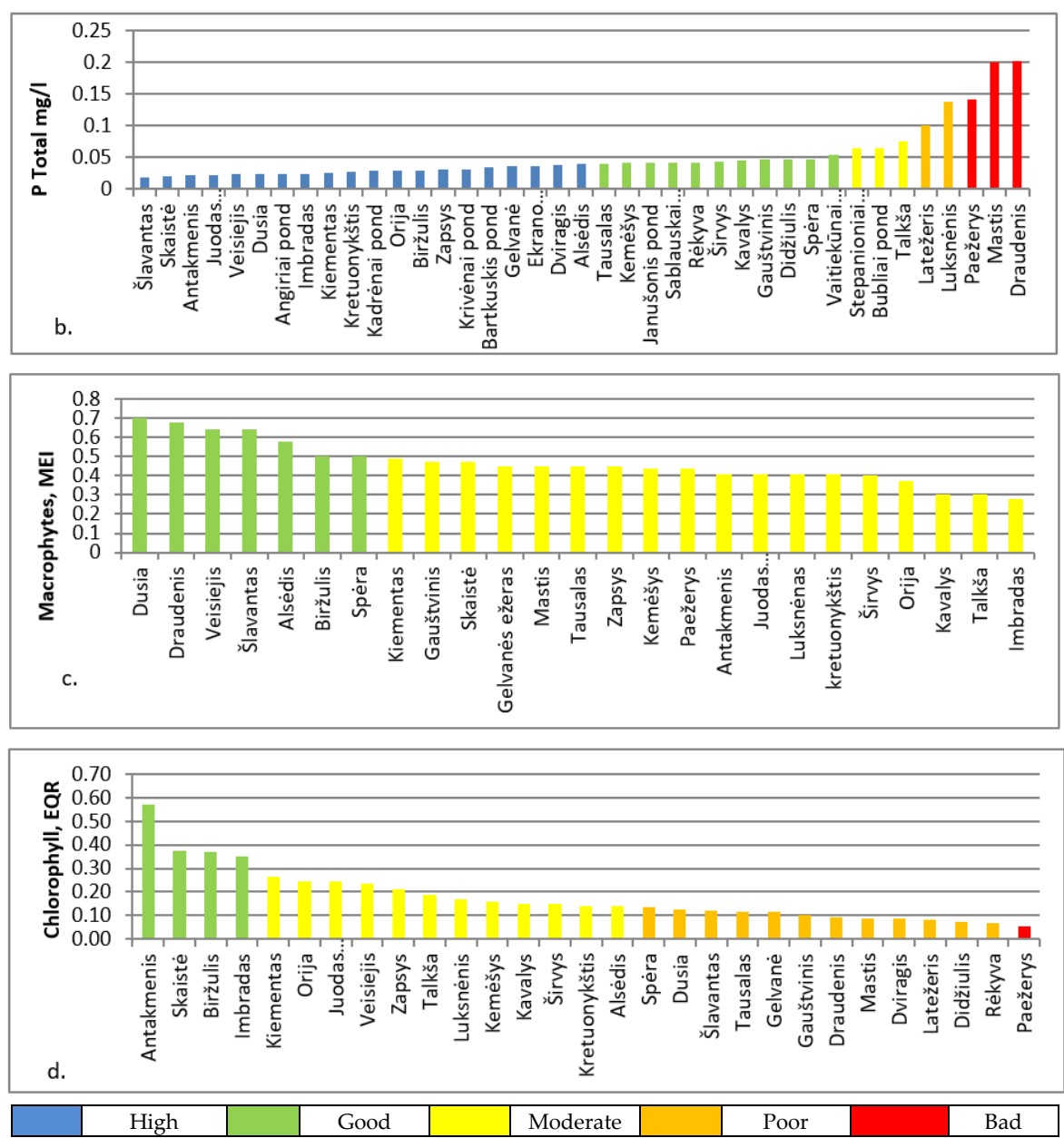

**Figure 2.** Ecological status of the assessed water bodies at risk: (**a**) N$_{total}$; (**b**) P$_{total}$; (**c**) macrophytes, MEI; and (**d**) chlorophyll, EQR. Ecological status.

The studied lakes do not meet the requirements of good and high ecological class on the following elemental data: 43.5 percent for total nitrogen, 20.5 percent for total phosphorus, 84.5 percent for chlorophyll a, and 62 percent for water bodies studied by macrophyte taxonomy.

*Relation of Hydromorphological Indicators with Hydrochemical and Hydrobiological Processes in Lakes*

The relationships between the water quality element values of water bodies and hydrometric indicators were evaluated. Water quality indicators are variables. Their change is dependent on the water depth, volume, sedimentation, overgrowth, metabolism time of water bodies, etc. Hydrochemical and hydrobiological processes occurring in the water mass are constantly closely related to hydromorphological indicators, the most important of which are water depth, volume and metabolism time. These are complex processes that are analysed insufficiently. However, it is obvious that the status of a deeper and larger volume of water deposits is better. Although the self-purification

of water is not precisely defined, research has estimated that greater water depth and volume in aquatic fauna functioning conditions and water quality indicators are better, as mentioned in [44].

Scatter diagrams (Figure 3) show that larger average and maximum lake depth indicates lower $P_{total}$, $N_{total}$ and macrophyte taxonomic composition values (better ecological status class), higher chlorophyll a EQR, ichthyophane taxonomic composition and higher values of LFI and the macroinvertebrates index of the Lithuanian lakes (better ecological status class). The linear regression equations developed can be used to calculate the correlation between the value of water quality indicators and water depth.

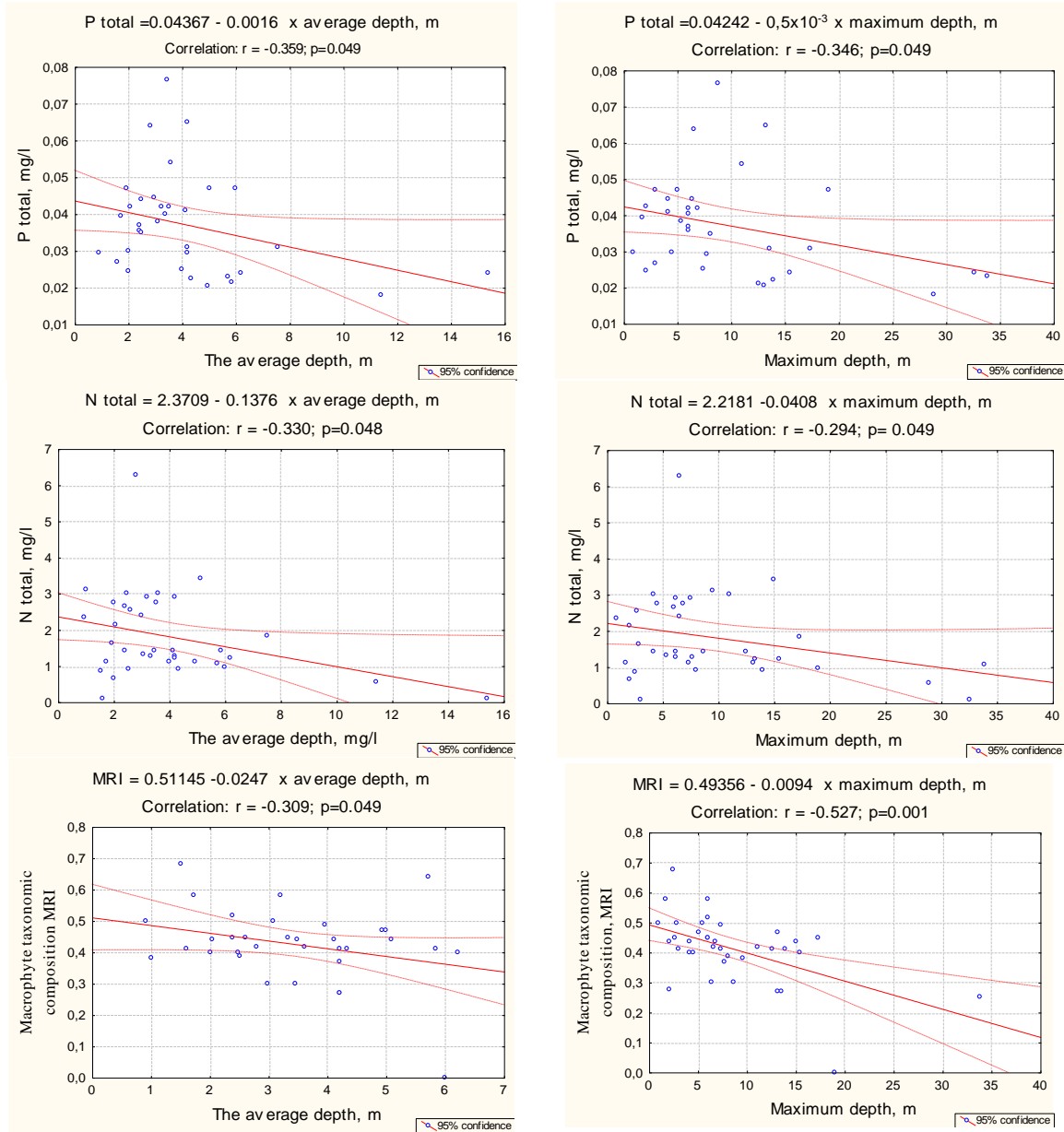

**Figure 3.** *Cont.*

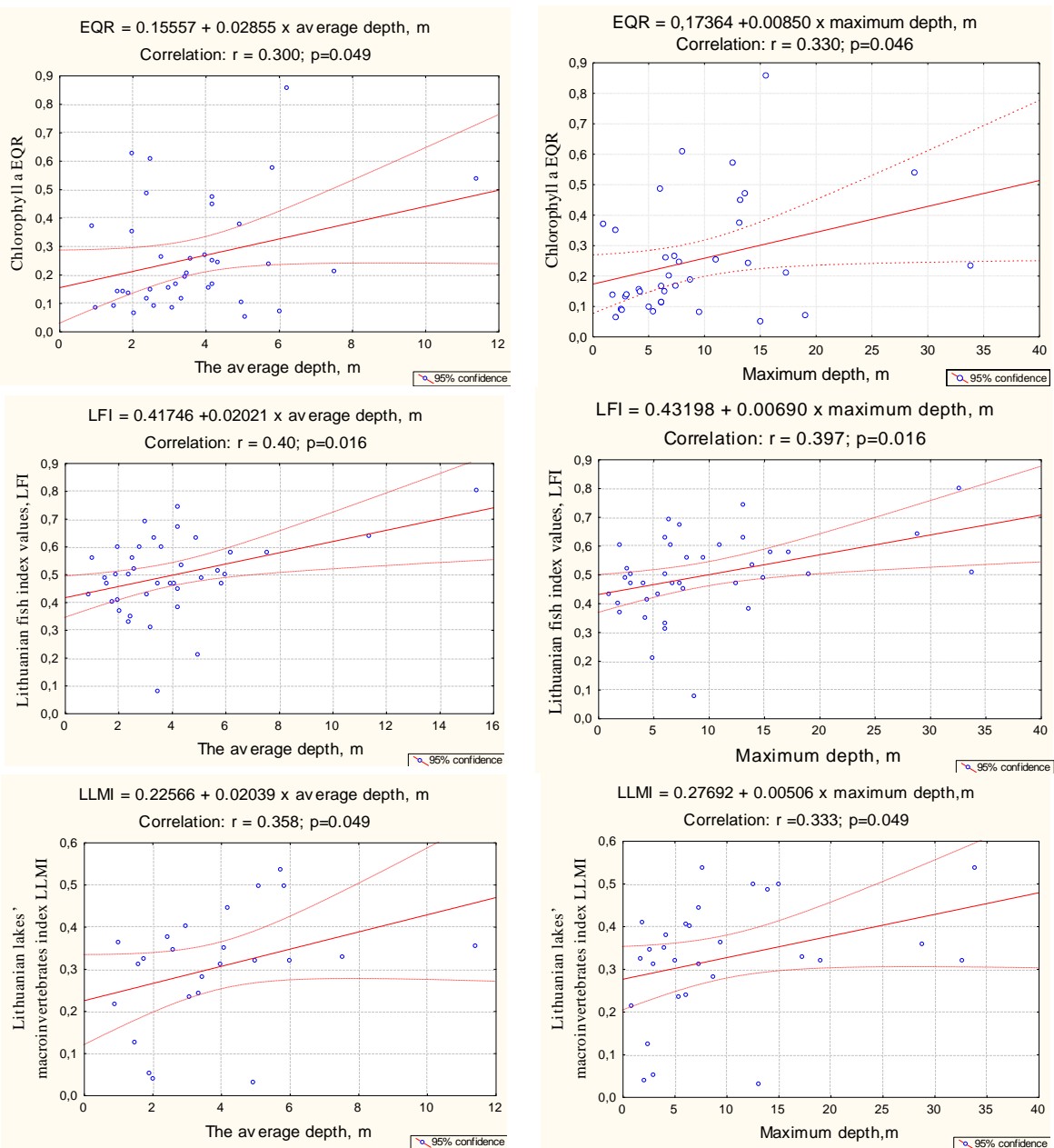

**Figure 3.** Water quality indicators (P$_{total}$; N$_{total}$; macrophyte taxonomic composition MRI, chlorophyll a EQR, ichthyophane taxonomic composition and higher values LFI and Lithuanian lakes' macroinvertebrates index LLMI) and their mean and maximum depths in the lakes assessed.

The research showed that increasing the depth of the lake, can improve the ecological status of the lake by all indicators measured [45].

Charts (Figure 4) show that larger lake areas have lower values of P$_{total}$ and N$_{total}$. This indicates a better water body ecological status class. Additionally, a larger lake area has a higher taxonomic composition of ichthyophane in LFI; a zoobenthos taxonomic composition indicator for LEMI and taxonomic composition of macrophyte MRI also indicate a better ecological status class. However, the larger the area of the lake is, the lower the chlorophyll a EQR values, indicating a worse ecological status. The results are presented in Figure 4.

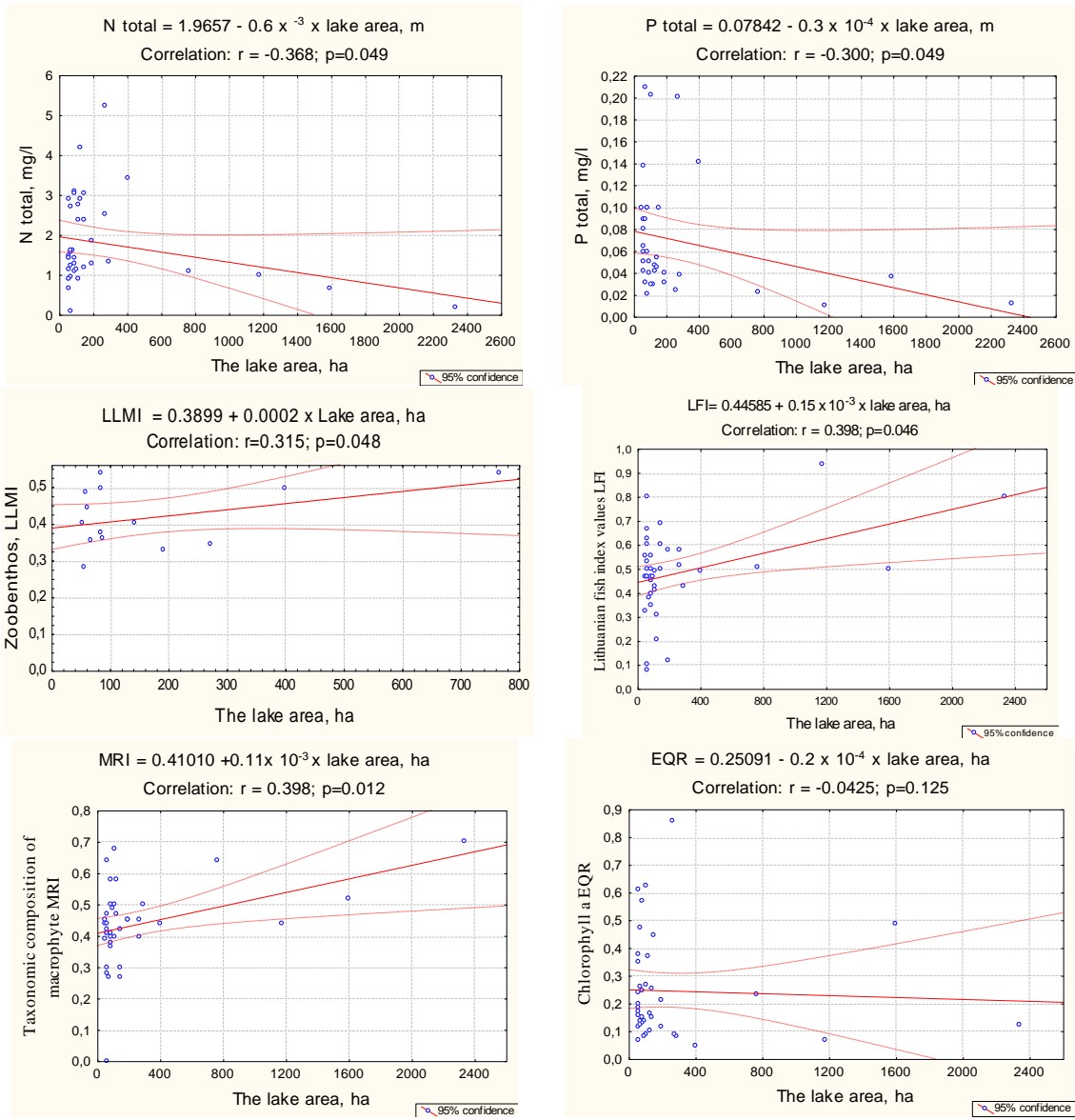

**Figure 4.** Water quality indicators ($P_{total}$; $N_{total}$; zoobenthos taxonomic composition indicator for LLMI, ichthyophane taxonomic composition and higher values of LFI and macroinvertebrates index, macrophyte taxonomic composition MRI, chlorophyll a EQR) and water area in the assessed lakes.

Linear regression equations created help calculate the values of the water quality indicators on varying lake areas.

The influences of water exchange rate in the pool on indicator values calculated. The results are presented in Figure 5.

Dissemination charts show rapid water exchange improvements in the conditions of the lake under nitrogen, phosphorus and chlorophyll a EQR values. Faster water exchange in the lake indicates lower $P_{total}$ and $N_{total}$ values and shows a good ecological status class of the water body. Faster water exchange in the lake also indicates higher chlorophyll a EQR values and shows a good ecological status class of the water body. Nonetheless, slower water exchange correlates with better ecological status of the macrophytic taxonomic composition of MRI, the ichthyofauna taxonomic composition, and Lithuanian lakes' macroinvertebrates index indicator of zoobenthos, indicating a better ecological status class of the water body.

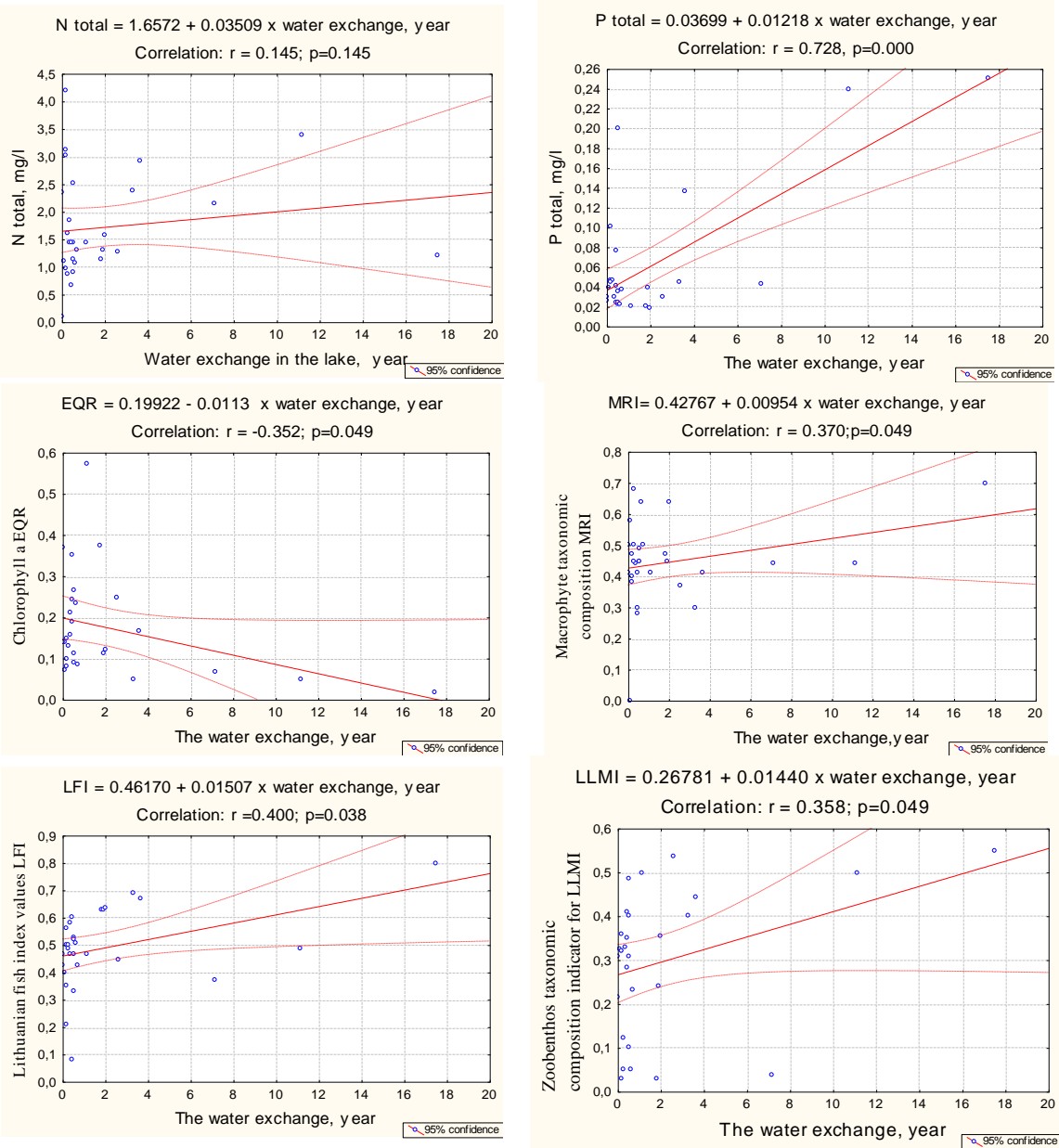

**Figure 5.** Water quality indicators ($P_{total}$; $N_{total}$; chlorophyll a EQR, macrophyte taxonomic composition MRI, ichthyophane taxonomic composition and higher values of LFI, zoobenthos taxonomic composition indicator for LLMI) and the water exchange in the assessed lakes.

## 4. Discussion

Changes in water quality indicators are much slower in large water bodies. The larger and deeper the water body is, the better the conditions for spontaneous water purification, slower hydrobiological processes, and slower accumulation of sediment. If the water body is shallow, hydrobiological and hydrochemical processes occur more intensely, organic sediments accumulate intensively, and the water body becomes overwhelmed and begins to disappear. The faster the water exchange, the lower the phosphorus concentration in the water and sludge [46]. Polish researchers have found that lakes with higher water temperatures have higher mean depth, maximum depth, and water transparency [47]. Water depth has an impact on the abundance and diversity of fish. Recent studies have shown that fish abundance and alpha diversity were positively correlated with wetland water depth. A. Garunkštis showed the remains of dead plankton, along with $CaCO_3$ crystals, slowly sinking into deeper layers.

Within a month, they only immersed approximately one metre. While sinking plankton residues oxidize and emit much $CO_2$, $O_2$ is reduced. With increased $CO_2$, deeper water layers dissolve $CaCO_3$ crystals and release $HCO^{-3}$, $Ca^{2+}$ and $Mg^{2+}$. As a result of these processes, the mineralization of water increases, and oxygen and pH levels decrease with depth. In shallow lakes and in the shallow areas of deep lakes, plankton remains are unable to oxidize, and $CaCO_3$ crystals are dissolved until they reach the bottom of the lake. Thus, the intensive accumulation of organic and carbonate deposits in shallow lakes and the shallow areas of deep lakes result in decreased mineralization of the water. The aforementioned processes also lead to rapid deposition in shallow lakes and ponds [48].

Because of geohydrological and climatic conditions, the eutrophication of lakes in natural conditions in Lithuania is low. Most lakes with wooded, sandy, not well-cultivated pools are mesotrophic. The nutrition of lakes with biogenic elements in a cultivated landscape is naturally increased because of the impact of agriculture on the environment. Therefore, in such natural conditions, one of the most common deteriorations of the status of water bodies is their sedimentation and the depletion of depth while not evaluating pollution inflow. The success of lake sludge treatment to improve water quality depends on the suspension of the inflow of external pollution loads [49]. In cases where the external pollution inflow charge has not been reduced, the impact of lake cleaning on water quality is negligible and short-lived [44].

One of the most important water quality indicators is the concentration of phosphorus. According to the available data of 40 investigated water bodies, the correlations between water depth and total phosphorus ($P_{total}$) and total nitrogen ($N_{total}$) concentrations are very reliable ($R^2 = 0.95$). As the average depth of the lake water increases, the concentration of total phosphorus in the lake decreases, and the ratio of total nitrogen concentration to average water depth is lower in comparison. The vertical phosphorus distribution demonstrates the trend of Ex-P, Fe-P, Oc-P Org-P, and TP values decreasing gradually with depth [50]. Many scientists have studied the impact of lake dredging on the lake's ecosystem [51–56]. The water quality of the lake can be improved by removing the surface layer of the seabed with a high concentration of phosphorus and controlling its access to the lake for P-limiting conditions for algal growth (e.g., [57–60]). Therefore, one of the most effective measures to improve the status of water bodies is to increase the depth of shallow, damp water bodies by removing accumulated sludge. In the literature, the treatment of lakes is controversial because it leads to deterioration of water quality indicators because of the sediment resuspension and lake ecosystem imbalance [61]. Zooplankton is particularly sensitive to lake cleaning. The composition and abundance of zooplankton communities fundamentally change, leading to less eutrophic species dominating [62]. The amounts of dissolved salts, electrical conductivity, nitrate nitrogen concentrations, phosphorus, organic matter, suspended solids, chlorophyll a concentration and water transparency increase significantly after purification. Therefore, the common view that a good lake ecological status can be achieved only by sludge treatment is wrong.

Based on the theoretical analysis [63], there are several problems that affect the efficiency of cleaning by reducing eutrophication in water bodies:

1. From a timely perspective, the bottoms of lakes are more a nutrient sediment than a source because there are opportunities for self-purification;
2. The consequence of organic matter degradation is the high concentration of nutrients on the surface of the sediment. In the sub-aquifer, a 'cloud of pollutants' occurs, which is agile and easily succumbs to resuspension, spreading to other layers of water;
3. The excretion of nutrients can occur only under certain conditions for a certain time. After cleaning, secondary pollution is reduced, but in time, after several months, it is back to the initial level before cleaning;
4. Cleaning efficiency depends on cleaning technology: sludge suction is superior to mechanical cleaning because with mechanical cleaning, the water is highly roiled;
5. Water quality issues (such as eutrophication) can be successfully solved by cheaper, properly selected alternative measures.

Many scientists claimed that dredging is not a panacea and that success is still a variable indicator [54,55,58,62]. Because of the high cost of dredging, studies on the removal of phosphorus from sediment layers must be carried out prior to dredging. There are only a few studies on the release of phosphorus (P) from the accumulated layer of sludge [53,64].

Lake cleaning can be continuous or partial. Cleaning the entire lake alters the average lake depth, whereby the type of lake is determined accordingly and should be evaluated before deciding to clean the lake because the ecological balance of the lake is damaged after the treatment and the indicators temporarily deteriorate. The impact on the ecosystem is lower when using partial cleaning; nonetheless, the impact on water quality is lower as phosphorus emancipation occurs from the remaining uncleaned sediments [65]. Deepening of the 1 ha/1 metre depth of a shallow lake to 4 metres would create conditions for self-cleaning of the water and a decrease in phosphorus concentration from 0.0784 mg/L to 0.0565 mg/L (0.0219 mg/L) (Table 2).

The sludge cleaning efficiency ranges from 0.005 to 0.44 cm per year. Sludge in the lake accumulates 0.12 mm per year on average.

Sludge removal from the water body is recommended when the average depth of the water body is less than 3.0 m, the thickness of the sludge layer is greater than the thickness of the water layer, and when possible signs of a problem are observed, e.g., annual algal blooming of lakes or ponds or parts of them and winter or summer fish suffocation. Prior to cleaning, complex botanical and zoological studies of the lake and the subdivision, a search for endangered species, research and/or monitoring of the hydrochemical status, and studies on the accumulated sludge layer and chemical co-operation must be assessed. After cleaning, an ecosystem characteristic becomes mesotrophic, the necessary species of macrophytes are planted, and the water body is stocked with predatory fish (pike, redfish, perch, catfish, etc.,). No less than 50 percent of the water body area should be cleaned. When cleaning by multiple stages, in the first stage, at least 10 percent of the water body should be cleaned when the water body area is larger than 50 ha and at least 20 percent if the water body area is less than 50 ha. At least 1.5 m of the sludge layer should be cleaned (from a coastal area to the mineral soil layer). The sludge is removed from the mineral soil layer or to a water depth of at least 4.0 m. The entire area of the water body risk factor is cleaned.

In reference to the results of natural conditions and modelling (WASP model), Yenilmez and Aksoy [67] found that purification of the entire lake area results in a short-term effect on the reduction in phosphorus concentrations in the absence of the interruption of external pollution. From a long-term perspective, partial cleaning combined with anti-pollution measures is more effective. This method is especially suitable for lakes that are in protected areas, and cleaning of the entire lake is not a suitable option, whereas cleaning the entire lake area has a negative impact on bottom invertebrates in the lake, and partial cleaning in the most densely populated areas is superior in this respect. In addition, this method is cheaper when compared to cleaning large lakes and has an obvious impact on the hydrochemical indicators (especially phosphorus).

**Table 2.** Estimation of the conditions for the purification of purified lakes (Source: [66]).

| The Name of the Lake | Sludge Cleaning (Year) | Depth after Cleaning | The Duration of New Sludge Accumulation from Purification (Years) | The Thickness of the Newly Formed Layer of Sludge, cm | | | Avg. Sludge Accumulation Intensity, mm/Year | Comments |
|---|---|---|---|---|---|---|---|---|
| | | | | Min | Max | Avg. | | |
| Druskonis | 1971–1973 | 3.5–4.3 | 34 | 3 | 19 | 12.9 | 0.38 | Four places with rain drain water leak from the streets. Black sediment. |
| Valdakio (Varėna dist.) | 1976–1978 | 3.5–4.0 | 29 | 1 | 7 | 1.5 | 0.005 | The lake is surrounded by mostly coniferous forests. Pink precipitate. |
| Mergelių akelių near Merkinė | 1978–1979 | 3.2–4.0 | 28 | 5 | 20 | 12.3 | 0.44 | Sediment from agricultural fields. Lake elusive. Greenish precipitate. |
| Ilgučio (Lygainių) Trakai district | 1978–1980 | 3.3 | 27 | 4 | 18 | 8.1 | 0.30 | A total of 100–150 m of the lake shore was cleaned between the villages of Lygainiai and Gudeliai |

## 5. Conclusions

When assessing 39 lakes and ponds from 2014 to 2018, the study showed that when measuring total nitrogen, only 56.5 percent of the lakes were assigned good environmental status; when measuring total phosphorus, 79.5 percent met the criteria. When measuring chlorophyll a status, 15.58 percent of the lakes met the criteria, and 38 percent met the criteria when measuring macrophytic taxonomic composition.

The study proved that higher maxima and average depths of lakes correlate with lower $P_{total}$, $N_{total}$ yield, and macrophyte taxonomic composition values, indicating higher ecological status class; higher chlorophyll a EQR, ichthyofauna taxonomic composition indicator for LFI and Lithuanian lakes' macroinvertebrate index (higher ecological class) values are.

Larger lake areas contain smaller amounts of $P_{total}$ and $N_{total}$, indicating better ecological status class. Higher ichthyophane taxonomic composition in LFI, zoobenthos taxonomic composition indicator for Lithuanian lakes' macroinvertebrates index (LLMI) and taxonomic composition of macrophytes MRI indicate better ecological status class. Larger lake areas contain lower chlorophyll a EQR values (worse ecological class status). Rapid water exchange improves the condition of the lake in addition to nitrogen, phosphorus and chlorophyll a EQR values. The faster the water exchange in the lake is, the lower the $P_{total}$ and $N_{total}$ values (better ecological class status); faster water exchange in the lake also means higher chlorophyll a EQR values (better ecological class status). However, slower water exchange indicates better ecological status of the macrophytic taxonomic composition of the MRI, the ichthyofauna taxonomic composition and the Lithuanian lakes' macroinvertebrates index indicator of zoobenthos.

In conclusion, one of the most effective measures to improve the status of water bodies is to increase the depth of shallow, damp water bodies by removing accumulated sludge.

**Author Contributions:** L.Č. designed the study and performed the experiments; L.Č, D.Š. and M.D. performed the experiments, analysed the data and wrote the manuscript. All authors have read and agreed to the published version of the manuscript.

**Funding:** This research was funded by the Polish National Agency for Academic Exchange under the International Academic Partnerships Programme. Publication is the result of participation in the project "Organization of the 9th International Scientific and Technical Conference entitled Environmental Engineering, Photogrammetry, Geoinformatics—Modern Technologies and Development Perspectives.

**Conflicts of Interest:** The authors declare no conflict of interest.

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
