# Peer review of "Relationship between the Water Quality Elements of Water Bodies and the Hydrometric Parameters: Case Study in Lithuania"

_water, doi:10.3390/w12020500_

Round 1

Reviewer 1 Report

Dear authors,

the manuscript was adopted, however I suggest to check the English by a native Speaker or professional language editing:

e.g. Figure 2:  why do you "invent" new terminology (average, very bad)? use the official WFD scaling (high, good, moderate, poor, bad) - see https://ec.europa.eu/environment/pubs/pdf/factsheets/water-framework-directive.pdf

After language editing, the manuscript can be accepted for publication.

Reviewer 2 Report

Revised according to the reviewer's comments. However, even if a significant correlation is obtained, the persuasiveness of the paper is very weak. I look forward to future research progress.

Author Response

Thank you for your hard work in improving the quality of our manuscript. We did the hard work of studying 29 lakes and 10 ponds. We may have lacked experience in data processing and analysis. We believe the topic is relevant not only in our country, but in others, so we will continue our research and try to improve the analysis.

This manuscript is a resubmission of an earlier submission. The following is a list of the peer review reports and author responses from that submission.

Round 1

Reviewer 1 Report

Dear Authors,

this is an important contribution related to the implementation of WFD, however some changes are required in order to improve the manuscript:

Please change the title, I suggest something like “Relationship between Biological Quality Elements of Water Bodies and Hydrometric Parameters: case study in Lithuania”.

Line 13: according to WFD “water bodies” include as well rivers! Thus be more specific here!

Line 15: of selected Lithuanian lentic water bodies

Line 25: delete “Human”

Line 36/37: in surface waters

Line 37: delete “fast”

Line 75: have been defined (not “isolated”)

Line 80/81: how many of those lakes are considered as a water body? Add information on the designation process (e.g. lakes bigger than XX ha).

Line 87 (wherefrom is the information from the 1st sentence?) + 90 (at the end of the sentence that begins with “one study”): citation missing!

Line 105/106: of selected Lithuanian lentic water bodies

Line 108-143: you could shorten the text and provide all the methodological aspects as a citation (and maybe add a link to the resource as well)

Line 135-139: up to now I thought the manuscript only covers lakes! Why do you mention DSFI? Also in Fig1 and Tab 1 I can only see lakes!

Table 1: where is the information from (citation missing)?!

Improve the quality of Figure 1!

Line 150-154: improve the text – it`s mainly methodology (and explain in a sentence what you want to say about “Water bodies at risk:”). Move 161-163 up à those are the real results!

Try to provide Fig 2 on one page! The legend should be above the caption!

Line 171: what about the factor “temperature”? can you include temperature? Also have a look to following publications (https://www.limnology-journal.org/articles/limn/abs/2019/01/limn180045/limn180045.html, https://onlinelibrary.wiley.com/doi/abs/10.1002/rra.3275) and include this aspect at least in the discussion…

Line 174: what do you mean with “more damaging”?

Line 176-178: this is methodology!

Line 195: “sludge treatment”? – please explain – e.g. suction dredging? Excavation?

Line 243-246: citations missing

Line 255: and what about the natural influence coming from peatlands / mires?

Line 274: also Olzhevsky tubes (e.g. https://www.zobodat.at/pdf/BERI_60_0183-0201.pdf, https://www.readcube.com/articles/10.2478%2Faep-2013-0004) have good results! Maybe to mention this in the discussion!

Divide the discussion in different sections, one devoted to the results of the study, another one dealing with the option “dredging” (4.1 – 4.2 - …), in order to improve the structure.

Line 297/298: what kind of dredging? Suction dredge? Excavation? ….

Line 308/309: is the table needed or can you just provide the citation in the text?

Reviewer 2 Report

This paper examines the relationship between water quality parameters for assessment of status under the Water Framework Directive and attempts to relate them to hydrometric and morphological parameters for a number of lakes in Lithuania.

There is a general structural problem throughout the entire manuscript. The information in the introduction is somewhat vague and does not seem to address many of the topics raised later in the manuscript including sludge management and there is some confused or poorly interpreted studies presented. The methods are poorly described and there is methodological detail presented in the results section. The results section does not explain the figures and tables presented in sufficient detail and the discussion section contains information which was not presented in the results nor seems to be related the aims of the manuscript or the results presented. In addition, the use of English is of a poor standard and requires significant revision. Although many of these issues could be addressed with major revisions, there are some additional concerns regarding the interpretation of some of the data. For these reasons I do not recommend that the manuscript be published in its current format. Please see detailed comments below for further explanation.

Line 20 : An explanation of the acronym LFI is needed.

Line 22: A full stop after the word ‘class’ is required to improve the sentence structure.

Line 27: A full stop after the word ‘values’ is required to improve the sentence structure.  

Line 36: Is anthropogenic activity only occurring in the surface water pool or throughout the catchment? This is not clear.

Line 38 – 39: Are the percentage of P and N figures relating to European studies or just Lithuanian studies? - I think this needs to be clarified.  

Line 45: It is not clear what is meant by ‘risk of failing to meet these objectives’. What risks?

Line 44 – Line 50: I think the description of the WFD could be improved – there is no explanation as to how chemical status and biological status are linked and what determines overall status. In addition, is ‘good’ status used here to refer to both ‘good’ and ‘very good’ status or is it just ‘very good’ status.

Line 53 – 56: The meaning of this section is not clear – what cases are the authors referring to – it should be clarified what study this is – are these figures based on combined information across many catchments or just one and if just one how is this likely to relate to the current study.

Line 66: What ‘feature change’ are the authors referring to? This paragraph is generally quite vague, not specifically clear what significant impacts were or what the hydrochemical or ecosystems changes were.

Line 20 – 80: Are these figures based on individual case studies, eg. Nutrient budgets for a lake – the implication from the way this is written is that for example 35% of P on the globe is in Estonia.

Line 76 onwards: Very unclear what lakes are included in WFD monitoring and how many lakes versus water bodies in general – so 1239 water bodies (I assume this includes rivers) – but there are 2850 lakes – so how many of these are part of the 1239 in WFD programme.

Line 88: Confusion over ‘at risk’ or ‘potentially at risk’ – surely 100%  of those classified as ‘at risk’ are also ‘potentially at risk?’ What was the point being made?

Line 102: repetition regarding point pollution – earlier discussion of point pollution, but now being referred to again.

Line 106: The aim fails to reflect the later discussion which largely focuses on lake management.

Line 108: Methods are poorly laid out, is there no original methodology that can be cited or a more detailed explanation of what was involved in some of the analyses than referring to national standards? No detail is given as to how N and P was analysed as an example. Detail about when samples were taken is given in the results section (Line 150), this is inappropriate. Samples were only taken in 2014 and 2017 and not during any other year, but this is not made clear in the methodology. An explanation should be provided as to why those particular lakes were included in the study. An explanation of the Chlorophyll a EQR is required for clarity.

Line 127: A heading for macrophytes, consistent with subheadings used earlier in the section is not provided. This should also be addressed for the macrozoobenthos and fish also.

Line 144: Figure of poor quality.

Line 147: Table 1 is not referred to in the text

Line 150 – 154: This is methodology and should not be included in the results.

Figure 2 is not described in detail in the text. A full explanation of results is required.

Line 162: It is not clear what the percentages refer to, the sentence states that the lakes surveyed do not classify as having good ecological status and then percentages are provided which suggest that many do, unless these percentages refer to something else. There is no detail provided as to what was found for each quality element, were there differences ?

Line 167: What is metabolism time? How is metabolism dependent on water depth sedimentation etc? Where is the reference for this statement or is this based on the results of this study? This section if very unclear, it is difficult to determine if what is presented is general information or based on results obtained in this study.

Line 176 – 178: This is for the Methodology and should not be presented here.

Line 186 – 188. Only one lake greater than 20 m in depth and without this data point it is unclear if the relationship would change.

Line 195: How does the research show that sludge treatment (which has not been described or mentioned in the manuscript until this point) improve the ecological status of a lake. There is no data either experimentally or observationally to show this in the manuscript. Is there data showing depth change as a consequence of sludge treatment? Is there data to show a separation between the effect of removing stored P (which is my understanding is one of the main reasons for removing bottom sediment as a management tool) as opposed the effect of increasing depth?

Line 206: Only three lakes in the dataset have an area greater than 400 ha, there does not appear to be any relationship with values obtained in lakes below this size. It is highly problematic to draw conclusions based on this data.

Line 213 – 218: Methods

Line 218: There was no explanation as to how water exchange was calculated for the study lakes in the methods section.

Line 236 – 246: It states that the status of lakes over 50 ha are better than smaller water bodies – I do not see this in the data presented, since no lakes in the study were under 50 ha according to Table 1.

Line 243: No reference provided for this study.

Line 245 – 254: Detailed information given on water chemistry in one of the study lakes – but none of this data is presented in this manuscript and no reference is provided for another study.

Line 256: The term ‘basin’ might be more appropriate here

Line 263: Again no reference.

Line 283: This is a strong statement regarding sludge treatment, given that no data regarding this topic has been provided in this manuscript. The detailed discussion of sludge treatment is not linked to the aims of the paper, the methodology or the results presented.

Line 336: Was there 29 lakes in this study (as previously mentioned) or 39?

Reviewer 3 Report

In recent years, research on water quality and ecosystem management in lakes has become very important in the context of global warming. This study discusses the relationship between water quality and hydrological parameters in several lakes in Lithuania. A large amount of measured water quality and hydrological data is collected and seems to contain valuable data, but the analysis method is extremely arbitrary and lacks objectivity. Therefore, unfortunately, this article is not recommended for publication in this journal. The following are examples of points that need to be corrected.

Figure 1 does not know where it is located on the world map. At a minimum, clear the place name and indicate the latitude and longitude. None of the relationships shown in Figures 3 and 4 seem to be highly correlated. If you are discussing correlations,you should show p-values etc. In the first place, the basis of the relationship between lake size, depth and water quality seems to be very weak. There is no explanation of the formula of each indicator.

That’s all